# Improving the Estimation of Gross Primary Productivity across Global Biomes by Modeling Light Use Efficiency through Machine Learning

Daqian Kong [1], Dekun Yuan [1], Haojie Li [1], Jiahua Zhang [2] , Shanshan Yang [1], Yue Li [3], Yun Bai [1] and Sha Zhang [1,*]

1 Space Information and Big Earth Data Research Center, College of Computer Science and Technology, Qingdao University, Qingdao 266071, China
2 Aerospace Information Research Institute, Chinese Academy of Sciences, Beijing 100094, China
3 School of Earth Science and Engineering, Hebei University of Engineering, Handan 056038, China
* Correspondence: zhangsha@qdu.edu.cn; Tel.: +86-150-6300-2891

**Abstract:** Estimating gross primary productivity (GPP) is important for simulating the subsequent carbon cycle elements and assessing the capacity of terrestrial ecosystems to support the sustainable development of human society. Light use efficiency (LUE) models were widely used to estimate GPP due to their concise model structures. However, quantifying $LUE_{max}$ (maximum light use efficiency) and representing the responses of photosynthesis to environmental factors are still subject to large uncertainties, which lead to substantial errors in GPP simulations. In this study, we developed a hybrid model based on machine learning and a LUE model for GPP estimates. This hybrid model was built by targeting LUE with a machine learning approach, namely multi-layer perceptron (MLP), and then, estimating GPP within a LUE model framework with the MLP-based LUE and other required inputs. We trained the hybrid LUE (H-LUE) model and then, compared it against two conventional LUE models, the vegetation photosynthesis model (VPM) and vegetation photosynthesis and respiration model (VPRM), regarding GPP estimation, using tower-based daily-scale observations from 180 flux sites that cover nine different plant function types (PFTs). The results revealed better performance ($R^2 = 0.86$ and RMSE = 1.79 gC m$^{-2}$ d$^{-1}$ on the test dataset) of the H-LUE model compared to the VPM and VPRM. Evaluations of the three models under four different extreme conditions consistently revealed better performance of the H-LUE model, indicating greater adaptability of the model to varied environments in the context of climate change. Furthermore, we also found that the H-LUE model can reasonably represent the responses of the LUE to meteorological variables. Our study revealed the reliable and robust performance of the developed hybrid LUE when simulating GPP across global biomes, providing references for developing better hybrid GPP models.

**Keywords:** light use efficiency; gross primary productivity; machine learning

## 1. Introduction

Gross primary productivity (GPP) is defined as the amount of organic carbon fixed by terrestrial green vegetation through photosynthesis per unit area per unit time [1]. It serves as the fundamental energy source for various physiological and ecological processes in vegetation and represents a crucial metric for carbon flux exchange between terrestrial ecosystems and the atmosphere. Accurate GPP estimation improves the simulation accuracy of subsequent carbon cycle elements (such as litter, soil respiration, etc.). In addition, it helps to accurately assess the capacity of terrestrial ecosystems to support the sustainable development of human society.

Over the last few decades, a series of terrestrial ecosystem models were developed to simulate vegetation primary productivity. Among them, satellite-based light use efficiency (LUE) models were widely used for estimating GPP due to their concise model structures. The widely used LUE models include the Carnegie–Ames–Stanford approach

(CASA) [2], the eddy covariance–light use efficiency (EC-LUE) model [3,4], the MOD17 algorithm [5], the vegetation photosynthesis model (VPM) [6], and the vegetation photosynthesis and respiration model (VPRM) [7]. In general, LUE models are constructed on the basis of two fundamental assumptions [5,8]. Firstly, they assume that the ecosystem GPP is linearly correlated with absorbed photosynthetically active radiation (APAR) through LUE, which is defined as the amount of carbon fixed per unit of APAR. Secondly, these models take into account that environmental stresses such as low temperature or water deficit may reduce the maximum light use efficiency ($LUE_{max}$) below its theoretical maximum value [9], where $LUE_{max}$ means the maximum light use efficiency of vegetation in the absence of any environmental stress. Thus, LUE models estimate vegetation GPP as $GPP = PAR \times FPAR \times LUE_{max} \times f$, where PAR refers to the incident photosynthetically active radiation per time period and FPAR denotes the fraction of PAR absorbed by the vegetation canopy, the product of which is absorbed photosynthetically active radiation (APAR); $LUE_{max}$ refers to the maximum light use efficiency and $f$ indicates the effects of environmental stresses on the maximum light use efficiency, the product of which is the actual LUE under real environmental conditions.

In LUE models, one of the most important parameters is $LUE_{max}$, which determines the accuracies of quantifying the magnitude of GPP. However, the determination of $LUE_{max}$ values remains largely uncertain. Firstly, $LUE_{max}$ is generally inversed using in situ GPP or net primary productivity (NPP) [10]. Therefore, differences in the used observations and differences in computing environmental stress on photosynthesis can lead to large differences in the $LUE_{max}$ values among the various LUE models [10,11]. The different $LUE_{max}$ values between various LUE models can result in significant differences in GPP estimates [12]. Secondly, taking into account the variability and uniqueness of various ecosystems in primary production, most current LUE models rely on look-up tables to determine vegetation specific model parameters including $LUE_{max}$ [5]. However, this parameterization strategy needs high-quality maps of vegetation cover. There are large uncertainties in current large area vegetation cover datasets, which can propagate into regional and global GPP estimates [11]. In addition, $LUE_{max}$ can also vary within a single biome type [13–15] and is a primary source of the uncertainty associated with GPP estimation [14]. Moreover, the differences in the descriptions and calculations of environmental stress between different LUE models can also cause differences in GPP simulations. Yuan et al. [16] found that the correlations of GPP simulations among seven LUE models were lower than that of simulated potential GPP without considering any environmental stress (i.e., $PAR \times FPAR \times LUE_{max}$), which indicated that the difference in the environmental stresses between different LUE models was one of the major sources of uncertainty in GPP simulation. Cai et al. [17] found that most of the LUE models they investigated found a strong positive correlation between the water availability and GPP estimates across large areas, different LUE models suggested that different areas exhibited this positive correlation, which indicates that different LUE models give varied GPP estimates due to varied projections of the ecosystem water balance.

The problem in estimating the LUE could be addressed by incorporating machine learning (ML) approaches, such as random forests (RF) and artificial neural network (ANN), which were applied to simulate various ecosystem processes [18–20]. Machine learning is powerful for dealing with large-scale datasets with multiple variables, especially when complex relationships exist among the predictors [21–24]. To decrease the uncertainties introduced by $LUE_{max}$ and environmental stresses in GPP simulations, we consider building a hybrid model based on ML and LUE models for GPP estimates. This hybrid model targets LUE with machine learning and then estimates the GPP within the LUE model framework with simulated LUE and other required inputs. LUE is affected by multiple environmental factors and has complex relationships with a series of variables [2,25–27], and thus, ML could potentially be useful for characterizing the variations in LUE across a wide range of environmental gradients and ecosystem types. Compared to the existing LUE models, such a hybrid model is expected to reduce the uncertainties introduced by $LUE_{max}$ and envi-

ronmental stresses in GPP simulations and improve GPP simulation accuracies by directly simulating LUE with ML. In fact, a number of studies applied machine learning methods to estimate purely empirical GPP models [25,28–30]. For example, Yang et al. [29] trained a support vector machine (SVM) to predict vegetation GPP using explanatory remote sensing variables, such as land surface temperature, enhanced vegetation index (EVI), land cover, and ground-measured climate variables. Based on support vector regression (SVR) and RF, Zhang et al. [30] first simulated a LAI time series directly using meteorological variables as inputs, and further modeled the GPP time series using modeled LAI time series and meteorological variables. However, compared to a hybrid model, purely empirical methods showed weaker adaptability to extreme environmental conditions [20].

The primary objectives of this study are to: (a) develop a hybrid LUE model based on ML and LUE models for GPP estimates; (b) compare the simulated results of the hybrid LUE model and two popular LUE models in order to identify the advantages of the hybrid LUE model over other commonly used LUE models; (c) evaluate the hybrid LUE model and two other LUE models under several extreme environmental conditions.

## 2. Materials and Methods

### 2.1. Data and Preprocessing

#### 2.1.1. Flux-Site Data

The daily-scale data from 180 flux sites from the FLUXNET2015 tier 2 dataset [31] were used for the analyses in the current study. These flux sites cover nine different plant function types (PFTs) including crops (CRO), deciduous broadleaf forests (DBF), evergreen broadleaf forests (EBF), evergreen needleleaf forests (ENF), grasslands (GRA), mixed forests (MF), savannahs (SAV), shrubs (SHR), and wetlands (WET). SHR include closed shrubs and open shrubs. A deciduous needleleaf forest site was included in ENF and woody savannah sites were included in SAV. The number of these flux sites among the 9 PFTs is as follows: CRO: 18, DBF: 21, EBF: 14, ENF: 42, GRA: 34, MF: 8, SAV: 14, SHR: 14, and WET: 15.

The data we used from the aforementioned dataset included seven meteorological factors, carbon dioxide mole fraction ($C_a$), and observed GPP data. Temporally continuous meteorological factors on a daily scale include air temperature ($T_a$), precipitation, incoming shortwave radiation ($R_g$), incoming longwave radiation ($R_L$), vapor pressure deficit (VPD), atmospheric pressure ($P_a$), and wind speed (WS). Considering the water storage capacity of the soil, we treated the accumulated precipitation value of 8 days including that day as the precipitation data of the day for model calculation, as described in a previous study [32]. The method to calculate the accumulated precipitation value of 8 days (P) on the nth day is as follows:

$$P = \begin{cases} PREC_1 + PREC_2 + \cdots + PREC_n, & 1 \leq n < 8 \\ PREC_{n-7} + PREC_{n-6} + \cdots + PREC_n, & n \geq 8 \end{cases} \quad (1)$$

where $PREC_n$ represents the precipitation on the nth day of a site. There are several GPP versions available in the dataset, among which we utilized the version labeled as 'GPP_NT_VUT_REF'. In this version, 'NT' denotes that the GPP was retrieved by partitioning the observed net ecosystem carbon exchange (NEE) applying the nighttime method [33], 'VUT' indicates that the data were filtered utilizing the friction velocity threshold (https://fluxnet.org/data/fluxnet2015-dataset/data-processing/, accessed on 21 May 2021), and 'REF' represents the most common reference value among those from 40 different FLUXNET2015 workgroups [34].

#### 2.1.2. MODIS Remote Sensing Data

In this study, six vegetation indices were employed, including normalized difference vegetation index (NDVI), enhanced vegetation index (EVI), normalized difference water index (NDWI), global vegetation moisture index (GVMI), normalized difference drought index (NDDI), and near-infrared reflectance for vegetation ($NIR_V$). These vegetation indices were calculated using the reflectance bands from the moderate resolution imaging

spectroradiometer (MODIS). The MODIS data used in this study correspond to the same years as the flux site data.

These reflectance bands were retrieved from the MODIS MCD43A4 Version 6 Nadir Bidirectional Reflectance Distribution Function (BRDF)-Adjusted Reflectance (NBAR) dataset (https://lpdaac.usgs.gov/products/mcd43a4v006/, accessed on 21 May 2021), which is generated daily using 16 days of Terra and Aqua MODIS data at a 500-m resolution. The retrieved data were processed using the following two steps to obtain daily records:

(a) Pixel values with snow or cloud cover were removed, i.e., only pixel values with a 'Reliability' band value of 0 or 1 were retained;

(b) For unavailable daily data, gaps were filled by linearly interpolating the closed available data over time.

### 2.1.3. Data Filtering

All the daily data were preprocessed to remove invalid records before the analyses. The procedure can be summarized as follows: First, the data before 2000 were excluded due to the unavailability of remote sensing data. Second, only the daily data with GPP aggregated from at least 50% of valid hourly samples within one day were used. Third, only the data periods with NDVI larger than 0.2 were selected to avoid the impact of non-growth season data [35].

### *2.2. Methods*

### 2.2.1. Basic Idea of Model Development

The model development process is shown in Figure 1. We developed a hybrid LUE model based on ML and LUE models for GPP estimates. The basic idea is that we first assimilate site-based GPP using an ensemble Kalman filter (EnKF) [36,37] to obtain the LUE. Second, we directly simulated LUE using multi-layer perceptron (MLP) driven by 15 variables. Third, we estimated the GPP within the LUE model framework using the modeled LUE and other required variables (i.e., PAR and FPAR). In the following sections, we provide a comprehensive account of the research method employed in this study.

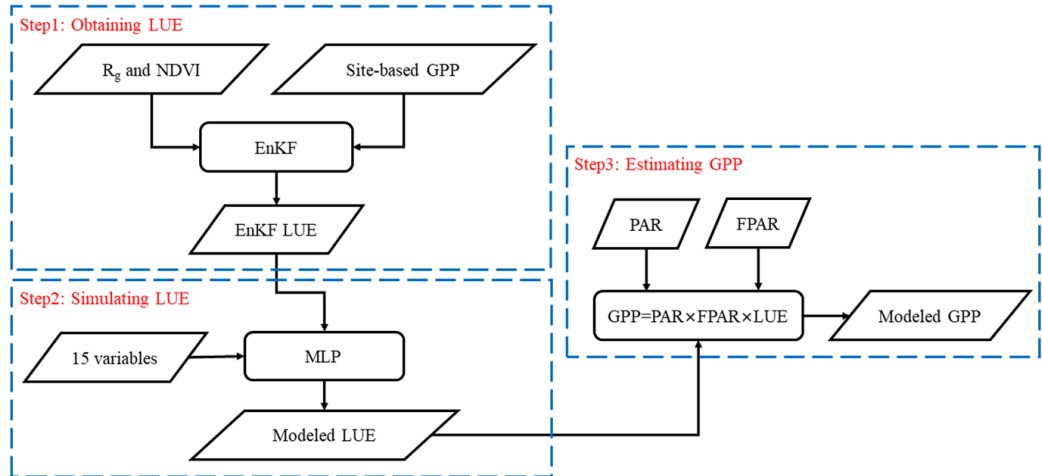

**Figure 1.** Flow chart of model development. For a description of the abbreviations and variables, please refer to Appendix A (Table A1).

### 2.2.2. Ensemble Kalman Filter

The Kalman filter (KF) is an algorithm that minimizes variance and updates the state estimate when measurements are obtainable [38]. The cost function of KF can be expressed as follows [39]:

$$J = \left(X_k^a - X_k^f\right)^T P^{-1}\left(X_k^a - X_k^f\right) + \left(Y_k - H\left(X_k^f\right)\right)^T R^{-1}\left(Y_k - H\left(X_k^f\right)\right) \tag{2}$$

where $X_k^a$ and $X_k^f$ denote the analyzed and forecast estimates, respectively, at time instant $k$, $Y_k$ denotes the vector of measurements, $H$ denotes the measurement operator that maps the model state $X_k - Y_k$, $P$ denotes the error covariance of the predicted model state, and $R$ denotes the measurement error covariance matrix. The KF update equation can be obtained by minimizing with respect to $X_k^a$ [37]:

$$X_k^a = X_k^f + K\left(Y_k - H\left(X_k^f\right)\right) \tag{3}$$

$$K = P_k^f H^T \left(HP_k^f H^T + R_k\right)^{-1} \tag{4}$$

where $P_k^f H^T$ denotes the cross covariance between a specific state and the prediction, $H\left(X_k^f\right)$, for an ensemble Kalman filter (EnKF).

$$P_k^f H^T = \frac{X_k^f - \bar{X}_k^f}{u - 1} q_k^T \tag{5}$$

$$q_k^d = H(X_k^f - \bar{X}_k^f) = (y_k^f - \bar{y}_k^f) \tag{6}$$

where $y$ is individual ensemble member of the prediction and $HP_k^f H^T$ denotes the error covariance matrix of the prediction.

$$P_k^f = \frac{(X_k^f - \bar{X}_k^f)(X_k^f - \bar{X}_k^f)^T}{u - 1} \tag{7}$$

$$HP_k^f H^T = \frac{q_k q_k^T}{u - 1} \tag{8}$$

where $u$ denotes the ensemble number.

We apply the EnKF to obtain the daily LUE through assimilating observations of GPP measured at flux sites. The estimated relative errors for observations are set to 15% according to the method described in a previous study [39]. The ensemble size is set to 200 [40]. The initial value of LUE is set to 0.075 gC J$^{-1}$ (d/s)$^{-1}$ ($\approx$0.87 gC MJ$^{-1}$). The lower limit of LUE is set to 0.01 gC J$^{-1}$ (d/s)$^{-1}$ ($\approx$0.12 gC MJ$^{-1}$), and the upper limit is set to 0.60 gC J$^{-1}$ (d/s)$^{-1}$ ($\approx$6.94 gC MJ$^{-1}$) [41].

### 2.2.3. Multi-Layer Perceptron

A multi-layer perceptron (MLP) was used to estimate LUE, based on 15 variables including PFT, $T_a$, accumulated precipitation value of 8 days (P), $R_g$, $R_L$, VPD, $P_a$, WS, $C_a$, NDVI, EVI, NDWI, GVMI, NDDI, and NIRV. All input variables except PFT were normalized to zero mean and one unit. For PFT, there are 10 possible types, i.e., C3 crops (CRO-C3), C4 crops (CRO-C4), DBF, EBF, ENF, GRA, MF, SAV, SHR, and WET, and we used dummy variables, which use the values 0 and 1 to represent a variable, as inputs to express them. We treated the LUE produced by EnKF as the target for model training. The MLP used here consists of a five-layer network (excluding the input layer): (1) an input layer directly connected to the input data, (2) four hidden layers with 30 neurons each, and (3) an output layer consisting of one neuron. The input layer serves to receive input data and transmit them to the next layer, the hidden layers are responsible for constructing the relationships between the input and output, and the output layer produces target estimates. We utilized the rectified linear unit (ReLU) as the activation function in our implementation of the MLP and employed early stopping as a means of preventing the overfitting of the model.

We first selected 12 flux sites that covered all 10 types of PFTs as a test dataset. Then, we shuffled the dataset of the remaining 168 flux sites randomly across time and sites and split it into two subsets: a training dataset and a validation dataset, accounting for 70% and 30% of the total number of flux sites, respectively. We repeated the training for several MLPs with an increasing number of hidden layers from 1 to 5 and an increasing number of neurons, e.g., 10, 20, 30, 40, and 50, to avoid overfitting. As a result, the MLP with four hidden layers and 30 neurons in each hidden layer is the best performing model in terms of root mean square error (RMSE).

### 2.2.4. Hybrid Light Use Efficiency Model (H-LUE)

To build a hybrid light use efficiency (H-LUE) model for GPP estimates, we implemented machine learning within the LUE model framework (Figure 2). We first simulated LUE using MLP driven by 15 variables. Then, we estimated GPP within the LUE model framework using simulated LUE, PAR, and FPAR. PAR for the H-LUE model is computed as follows.

$$PAR = 0.5 \times R_g \tag{9}$$

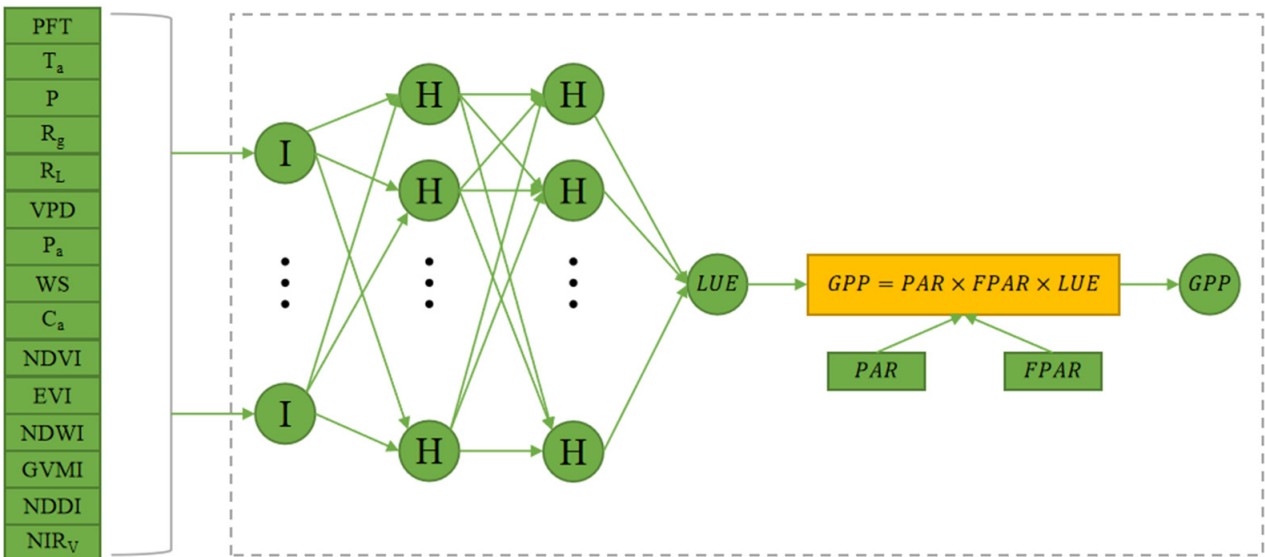

**Figure 2.** Flow chart of the H-LUE model. For a description of the abbreviations and variables, please refer to Appendix A (Table A1).

The value of 0.5 used in Equation (9) is an empirical approximation of the ratio between PAR and $R_g$ [42]. FPAR is calculated according to the empirical linear regression function [43]:

$$FPAR = 1.215 \times NDVI - 0.16 \tag{10}$$

### 2.2.5. Evaluating the Performance of the Models

We used daily observed GPP from flux sites to evaluate the performances of the GPP models. We employed commonly used evaluation metrics, including coefficient of determination ($R^2$), RMSE, and mean bias [44], to assess the performance of the GPP models.

### 2.2.6. Comparing the H-LUE Model with VPM and VPRM

To verify the advantages of the H-LUE model, we also ran two widely used LUE models, the VPM [6] and the VPRM [7], on the test, training, and validation datasets, for comparison. Detailed information regarding the two models can be found in Appendix B. VPM is a popular LUE model that was widely adopted in various studies. Many studies proved that this model has relatively high simulation accuracy among GPP models [45–47]. Additionally, the VPRM starts from the VPM. Therefore, these two models were used for comparison. The only parameter of VPM is $LUE_{max}$, and VPRM requires two parameters,

i.e., $LUE_{max}$ and $PAR_0$. For VPM, the $LUE_{max}$ values calibrated by Yuan et al. [11] were used for all PFTs except crops, and the $LUE_{max}$ values optimized by Bai et al. [42] for C3 and C4 crops, respectively, were used for crops. For VPRM, the parameter values calibrated by Yuan et al. [11] were used for all PFTs.

### 2.2.7. Assess the GPP Models under Extreme Environmental Conditions

Furthermore, we compared the capacity of the H-LUE model and the two LUE models (VPM and VPRM) to estimate GPP under extreme environmental conditions. The comparisons were carried out on the test dataset. For extremely wet conditions, we investigated data of the 90th–100th percentile P; for droughts, we investigated data of the 0th–10th percentile P; and for heat waves, we investigated data of the 90th–100th percentile $T_a$. When $T_a$ was lower than $-5\,^\circ C$, the photosynthesis of plants almost completely stopped (see Section 4.4). Therefore, for cold waves, we investigated data with $T_a$ of $-5$–$5\,^\circ C$. The distribution of data for extreme environmental conditions is shown in Figure 3.

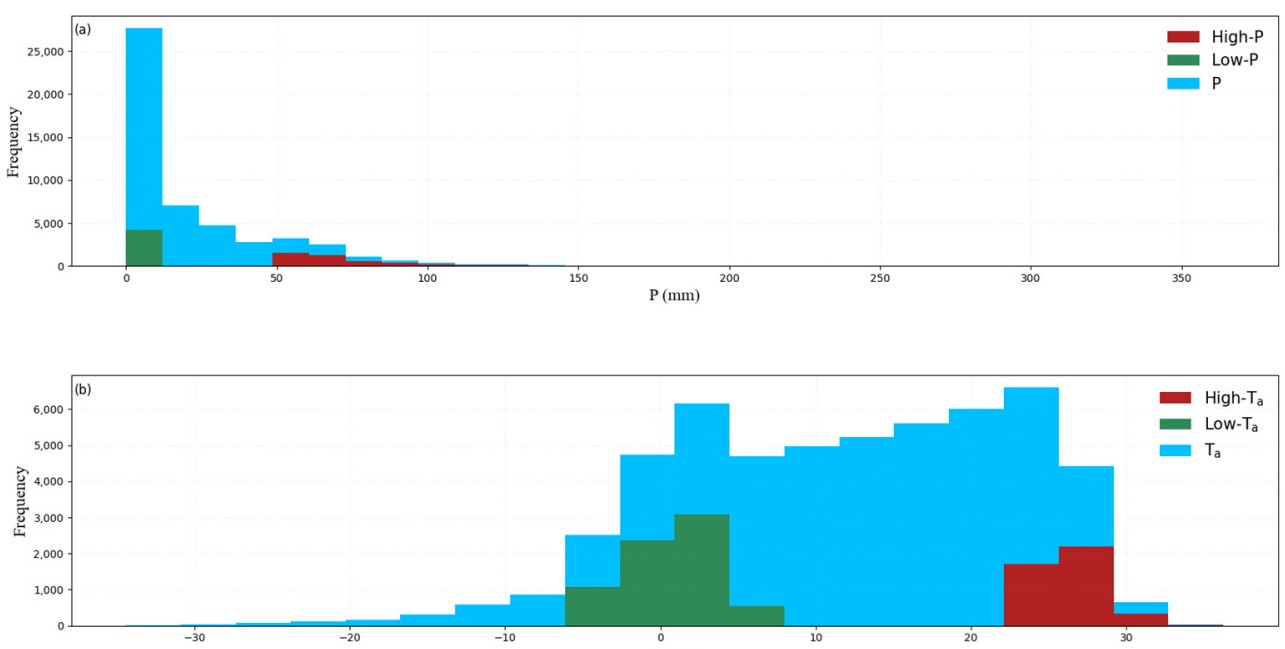

**Figure 3.** The distribution of data for extreme environmental conditions. (**a**) The distribution of data for extremely wet and dry conditions. (**b**) The distribution of data for high-temperature and low-temperature conditions. High-P, Low-P, High-$T_a$, and Low-$T_a$ denote extremely wet, dry, high-temperature, and low-temperature conditions, respectively.

## 3. Results

### 3.1. Evaluations of Three Models in Modeling LUE

Figure 4 presents the regression analysis between modeled LUE and LUE obtained by EnKF for the test dataset. For VPM and VPRM, modeled LUE is the product of $LUE_{max}$ and $f$. The results suggest that the MLP is capable of accurately capturing the variations in LUE obtained by EnKF with an $R^2$ value of 0.60 for the test dataset. MLP also produced a low RMSE of 0.36 gC $MJ^{-1}$ for the test dataset. On the test dataset, MLP performed significantly better than VPM ($R^2 = 0.38$ and RMSE = 0.43 gC $MJ^{-1}$) and VPRM ($R^2 = 0.17$ and RMSE = 0.69 gC $MJ^{-1}$). MLP also showed better performance than VPM and VPRM on the training and validation datasets (Appendix C, Figure A1). The results suggest that MLP can reduce the uncertainties in quantifying LUE as compared to VPM and VPRM. In other words, MLP can indeed reduce the uncertainties introduced by $LUE_{max}$ and environmental stress factors of conventional LUE models.

### 3.2. Evaluations of Three Models in Modeling GPP

The performances of the H-LUE model and the two LUE models in estimating the GPP of the test dataset are summarized in Figure 5. The results demonstrate that on the test dataset, the H-LUE model yields significantly higher $R^2$ and lower RMSE ($R^2 = 0.86$ and RMSE = 1.79 gC m$^{-2}$ d$^{-1}$) values as compared to VPM ($R^2 = 0.73$ and RMSE = 2.79 gC m$^{-2}$ d$^{-1}$) and VPRM ($R^2 = 0.68$ and RMSE = 3.26 gC m$^{-2}$ d$^{-1}$). The H-LUE also performed better than VPM and VPRM on the training and validation datasets (Appendix C, Figure A2). The results emphasize that combining the LUE model with machine learning can lead to improved performance compared to conventional LUE models.

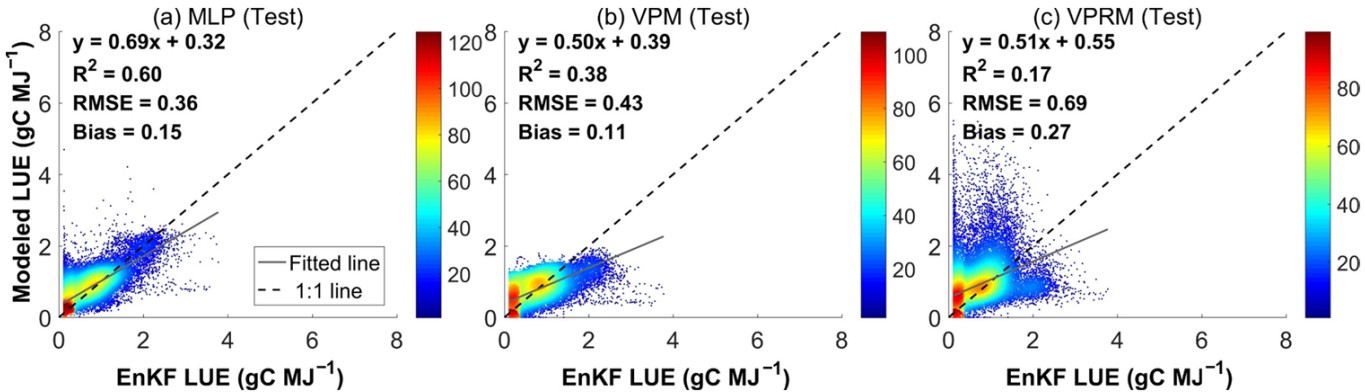

**Figure 4.** Modeled light use efficiency (LUE from MLP, VPM, and VPRM) vs. LUE obtained by EnKF for the test dataset. The color bars represent the density of points within a circle area with a radius of 0.25 gC MJ$^{-1}$.

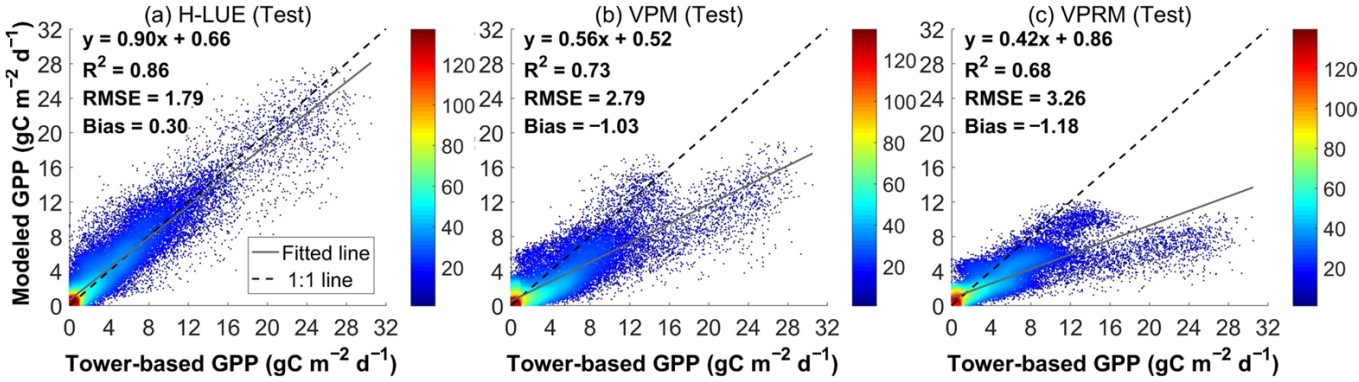

**Figure 5.** Evaluation of three models (H-LUE, VPM, and VPRM) when estimating daily gross primary productivity (GPP) on the test dataset. The color bars represent the density of points within a circle area with a radius of 1.0 gC m$^{-2}$ d$^{-1}$.

### 3.3. Performances of H-LUE in Different PFTs

We tested the performances of the H-LUE model in different PFTs on the test dataset, and the results are shown in Figure 6. The H-LUE showed significant differences in performance among different PFTs. The best model performance was observed in C3 crops, C4 crops, and deciduous broadleaf forests, with an $R^2$ value of 0.92. However, for evergreen broadleaf forests, savannahs, and shrubs, the H-LUE showed low performance and captured only small variations in GPP with $R^2$ values of 0.43, 0.37, and 0.51, respectively. Additionally, the H-LUE showed intermediate performance in the remaining four PFTs (i.e., evergreen needleleaf forests, grasslands, mixed forests, and wetlands). In general, the H-LUE showed satisfactory performance across PFTs. However, poor performance in a few vegetation types should also be noted.

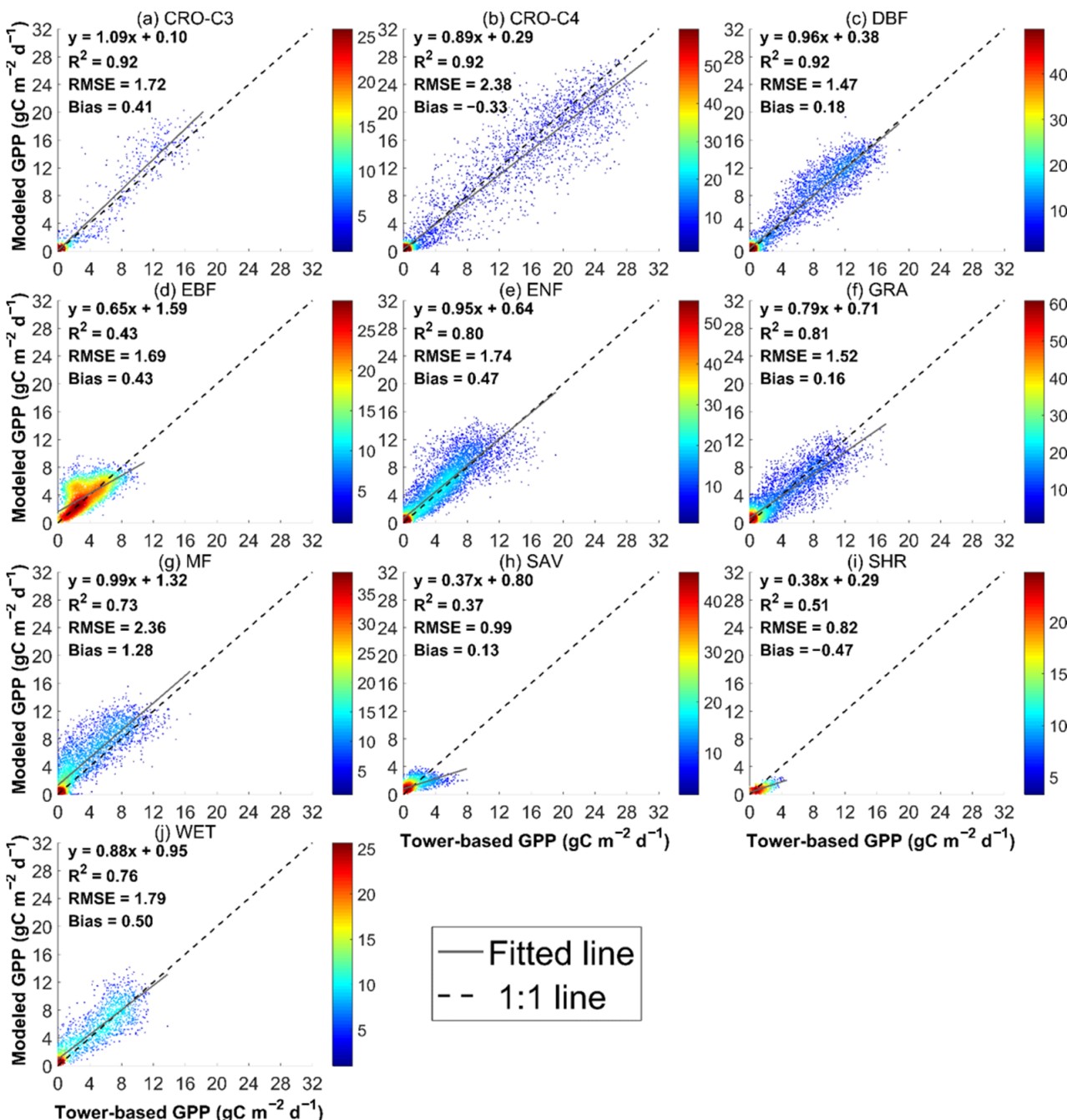

**Figure 6.** Performance of the H-LUE model when estimating daily GPP in different PFTs. CRO-C3: C3 crops; CRO-C4: C4 crops; DBF: deciduous broadleaf forests; EBF: evergreen broadleaf forests; ENF: evergreen needleleaf forests; GRA: grasslands; MF: mixed forests; SAV: savannahs; SHR: shrubs; WET: wetlands. The color bars represent the density of points within a circle area with a radius of $1.0 \text{ gC m}^{-2} \text{ d}^{-1}$.

### 3.4. Evaluations of Three Models under Extreme Conditions

We assessed the performances of the three GPP models under four types of extreme environmental conditions, as we clarified in Section 2.2.6. The performance comparison of the three GPP models is shown in Figure 7. For extremely wet (High-P), extremely dry (Low-P), and extreme high-temperature (High-$T_a$) conditions, the H-LUE model showed significantly better performance, with $R^2$ (RMSE) = 0.89 (2.28 gC m$^{-2}$ d$^{-1}$), 0.74 (1.42 gC m$^{-2}$ d$^{-1}$), and 0.90 (2.49 gC m$^{-2}$ d$^{-1}$), respectively, as compared to VPM and VPRM. For the extremely low-temperature (Low-$T_a$) condition, the H-LUE model yielded notably larger $R^2$ and

comparable RMSE values compared to VPM and VPRM. It should be noted that under the Low-$T_a$ condition, the ecosystem photosynthesis rate becomes very small, which explains the comparable RMSE between the three GPP models under such conditions. However, the H-LUE model tends to capture more variations in GPP under the Low-$T_a$ condition. These results suggested notable advantages of the H-LUE model over conventional LUE models under extreme conditions.

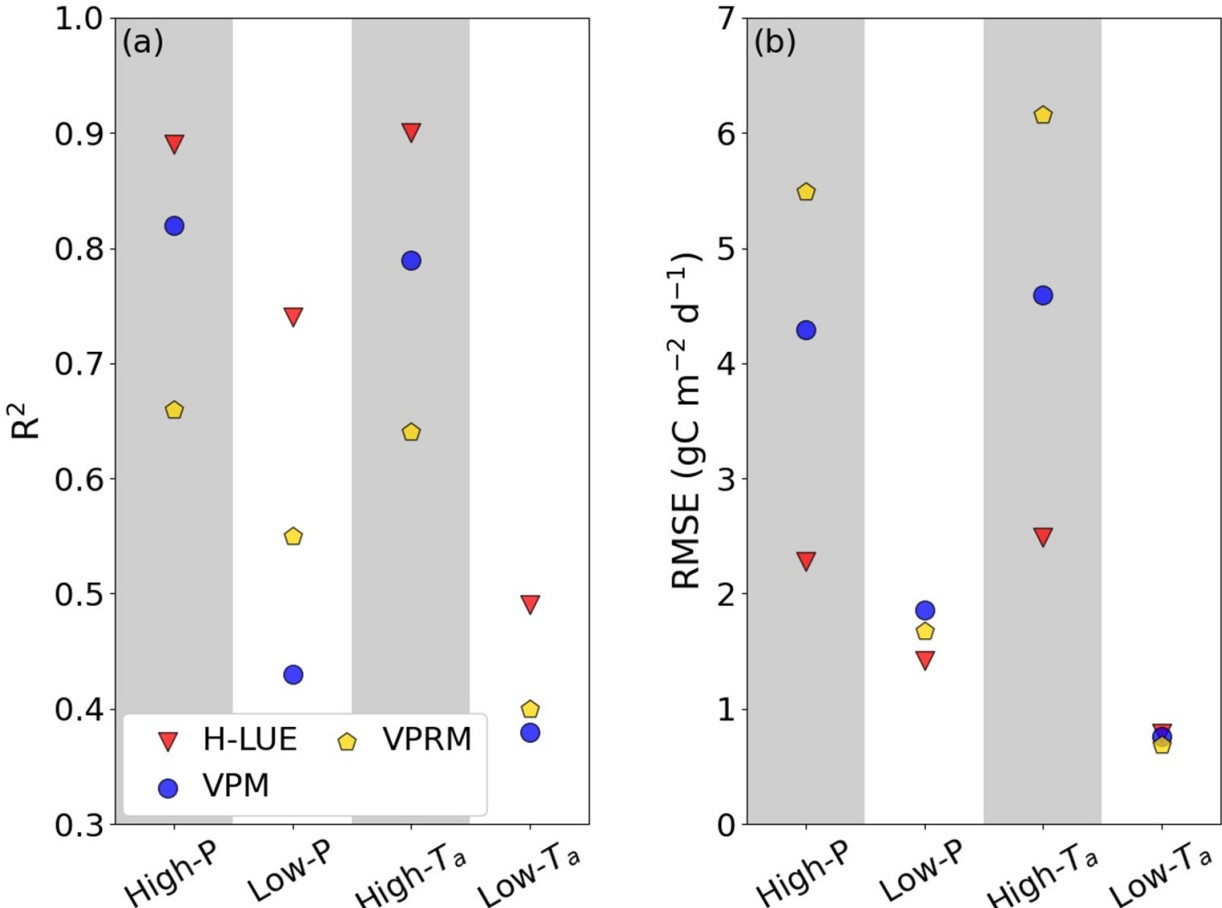

**Figure 7.** Comparison of performance metrics (**a**) $R^2$ and (**b**) RMSE among three models for modeling GPP in four extreme environmental conditions. High-P, Low-P, High-$T_a$, and Low-$T_a$ on the horizontal axis denote extremely wet, dry, high-temperature, and low-temperature conditions, respectively. The comparisons were carried out on the test dataset.

## 4. Discussion

### 4.1. Analyses of Bias

For the test dataset, the H-LUE model was able to simulate GPP accurately with a low bias of 0.30 gC m$^{-2}$ d$^{-1}$ (Figure 5a). Both VPM and VPRM underestimated GPP on the test dataset. VPM exhibited a large underestimation of GPP for the test dataset (Bias = −1.03 gC m$^{-2}$ d$^{-1}$) (Figure 5b), as did VPRM (Bias = −1.18 gC m$^{-2}$ d$^{-1}$) (Figure 5c). However, in the regression analysis between the modeled LUE and the LUE obtained by EnKF, the LUE modeled by VPM had a positive bias on the test dataset (Bias = 0.11 gC MJ$^{-1}$) (Figure 4b), and the LUE modeled by VPRM also had a positive bias on the test dataset (Bias = 0.27 gC MJ$^{-1}$) (Figure 4c). Both VPM and VPRM tended to overestimate the LUE when it was small and to underestimate the LUE when it was large (Figure 4b,c). The MLP can simulate LUE accurately with a low bias of 0.15 gC MJ$^{-1}$ for the test dataset (Figure 4a).

The photosynthesis capacity of plants In the middle of the growing season is much larger than that in the early growing season, while the conventional LUE models do not

account for the change in photosynthesis capacity and run with a constant $LUE_{max}$ value at all times. This is probably the reason why VPM and VPRM overestimate the LUE when is small and underestimate it when it is large. Meanwhile, the $LUE_{max}$ based on machine learning in the H-LUE is likely to be variable. In addition, VPM and VPRM use the land surface water index (LSWI) to account for the impacts of water stress on plant photosynthesis. However, a previous study [48] revealed that although the LSWI was sensitive to soil moisture variability, there was only a weak correlation between LSWI and GPP. This also reduces the accuracy of the GPP estimates of VPM and VPRM. The H-LUE model simulates LUE using multiple input variables (i.e., PFT, $T_a$, P, $R_g$, $R_L$, etc.), without using a constant $LUE_{max}$ or only two or three environmental stresses defined by a single LUE model, effectively avoiding the issues mentioned above.

VPM and VPRM exhibited a significant underestimation of the GPP, but not of LUE, which may be due to the relatively low FPAR used in these two models. Yuan et al. [10] found that among several LUE models, VPM had a relatively low FPAR. The FPAR used in VPRM is calculated using the same method as in the VPM, resulting in a relatively low FPAR as well.

*4.2. Comparison of H-LUE with Other LUE Models*

In order to further verify the performance of H-LUE, we ran another widely used LUE model (i.e., EC-LUE model) for additional comparison. Detailed information regarding the EC-LUE model can be found in Appendix B. We ran the four models (i.e., H-LUE, EC-LUE, VPM, and VPRM) on 11 sites of the test dataset (one site of the test dataset was excluded due to the unavailability of the net radiation data that is required by EC-LUE). The performance comparisons are shown in Figure 8. The H-LUE showed the best performance ($R^2 = 0.87$ and RMSE = 1.72 gC $m^{-2}$ $d^{-1}$) among the four models, which further confirmed the advantage of the H-LUE.

In addition, EC-LUE achieved higher $R^2$ and lower RMSE values when compared to the other two LUE models (i.e., VPM and VPRM), which was consistent with a previous model evaluation [16]. Yuan et al. [16] assessed the performance of seven LUE models (CASA, cFix, cFlux, EC-LUE, MODIS-GPP, VPM, and VPRM) using flux measurements from 157 eddy covariance sites and found that EC-LUE showed better performance than VPM and VPRM. These differences were expected. Model structural differences were considered the most important reason of different GPP estimates among models [16]. EC-LUE uses the same equation as VPM and VPRM to estimate temperature stresses, but the water stress equations are different. In the EC-LUE model, water stress is estimated using the ratio of latent heat flux to net radiation because decreasing amounts of energy partitioned to evaporate water suggests a stronger moisture limitation [49,50]. VPM and VPRM use a satellite-derived water index (LSWI) to estimate the seasonal dynamics of water stress [6]. A previous study indicated that water stress algorithms generate greater variation among models than temperature factors [16]. Another study revealed that although most models, including EC-LUE, VPM and VPRM, found strong positive correlations between the water availability and GPP estimates across large areas and various ecosystems, different LUE models showed different areas displaying this positive correlation [17]. Therefore, the difference in GPP estimates may be attributed to the model discrepancy in quantifying water stress. Moreover, FPAR of EC-LUE is calculated using NDVI, while that of VPM and VPRM is derived from EVI. Although NDVI and EVI are complementary vegetation indices [51], there was a significant difference in the long-term change between the two vegetation indices over the global scale [17].

It remains difficult to characterize the effects of environmental stresses such as temperature and water availability on vegetation photosynthesis over large areas from physical modeling, and this limits the accuracy of conventional LUE models. The best performance of H-LUE among the four models indicated that applying ML to simulate the LUE can improve the accuracy of estimating the GPP.

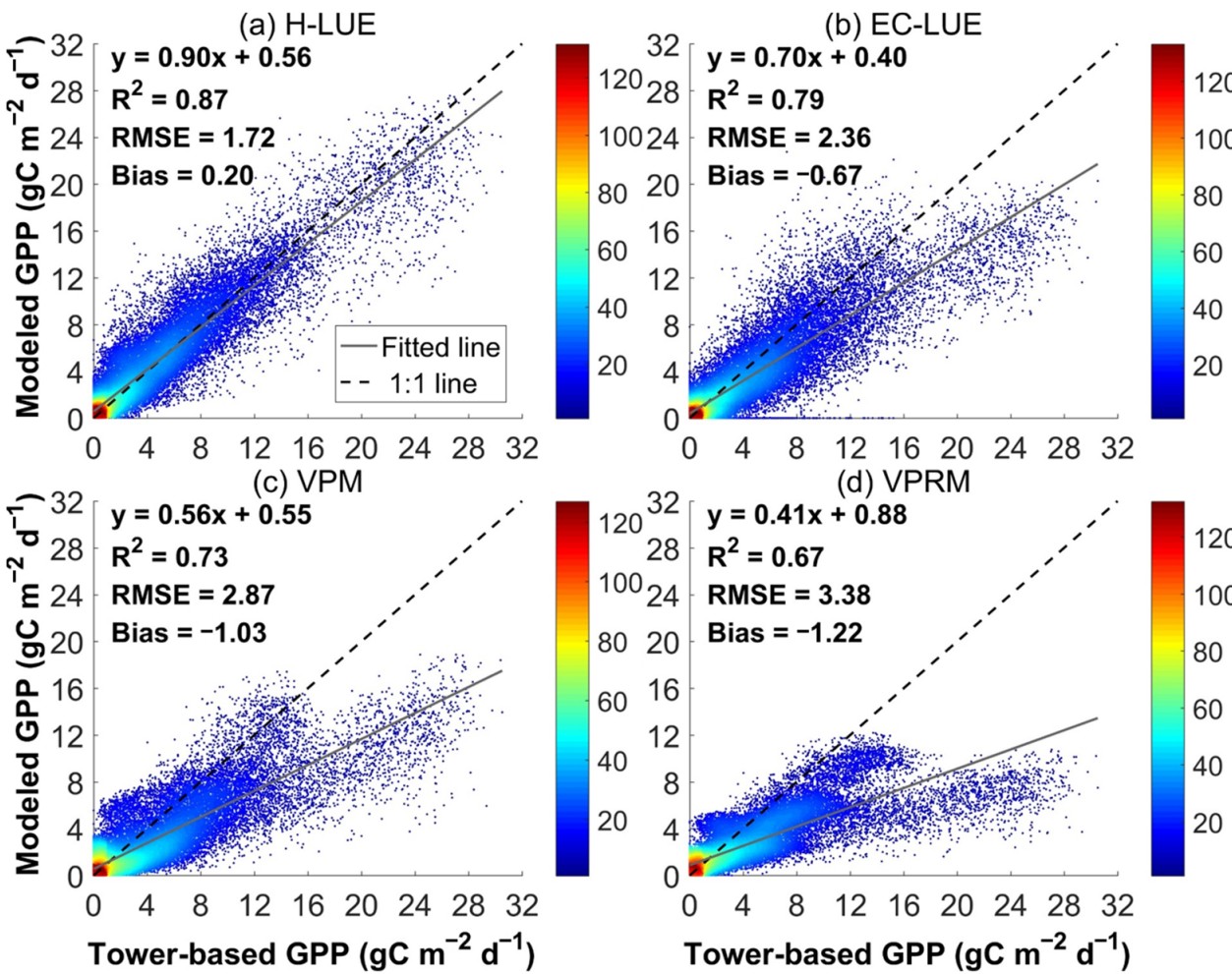

**Figure 8.** (**a**) H-LUE, (**b**) EC-LUE, (**c**) VPM, and (**d**) VPRM estimations of daily gross primary productivity (GPP) on 11 sites of the test dataset. The color bars represent the density of points within a circle area with a radius of 1.0 gC m$^{-2}$ d$^{-1}$.

*4.3. Different Performances of H-LUE in Different PFTs*

We found statistically significant differences in the performance of the H-LUE among PFTs (Figure 6). The H-LUE showed good performance for crops and deciduous broadleaf forests, and poor performance for evergreen broadleaf forests, savannahs, and shrubs. This conclusion is supported by previous studies [16,52]. Yuan et al. [16] assessed the performance of seven LUE models in six ecosystem types and found that all seven models showed good performance for deciduous broadleaf forests and poor performance for shrublands and evergreen broadleaf forests. Based on 17 models against observations from 36 North American flux towers, Raczka et al. [52] revealed that the models performed best for deciduous broadleaf sites, but not well for evergreen sites. In general, deciduous broadleaf forests show distinct seasonal dynamics of leaf phenology, and satellite data can accurately capture the phenology change [16], which is beneficial to GPP modeling. Meanwhile, evergreen broadleaf forests show subtle changes in the seasonal leaf phenology, and various environmental variables jointly determine the vegetation photosynthesis, which increases the difficulty of modeling [6].

*4.4. The Responses of LUE to $T_a$, $R_g$, and VPD in the H-LUE Model*

Whether in conventional LUE models or in the H-LUE model developed in this study, meteorological factors are essential input data that play a crucial role in ensuring the

accuracy of GPP simulation. The MLP can accurately model LUE and, therefore, can be applied to systematically understand the responses of LUE to meteorological factors.

Figure 9 shows the density plot of the LUE modeled by MLP with the three meteorological factors $T_a$, $R_g$, and VPD for the test dataset. As expected, the LUE increased sharply as $T_a$ increased, especially when $T_a$ was higher than $-5\ ^\circ$C. When $T_a$ was lower than $-5\ ^\circ$C, the LUE was very low and changed slowly, which indicated the inhibitory effect of low temperatures on plant photosynthesis. In general, chill could induce stomatal closure, precipitating a decline in photosynthesis [53]. LUE decreased with VPD, which indicates that high VPD can inhibit photosynthesis. An increase in VPD results in an enhanced transpiration rate, and stomata then respond by partially closing to reduce the water loss of plants [54], which leads to a decline in photosynthesis. As the light level increases, the canopy LUE decreases because of the light saturation effect [55]. However, the LUE showed insignificant responses to $R_g$ variations (Figure 8b), likely because other environmental factors can be covarying at the same time—for instance, $R_g$ can increase during reduced cloud cover while temperature can also increase. Moreover, previous studies found that an increased fraction of diffuse radiation during cloudy days enhanced plant photosynthesis [55,56]. Therefore, the variations of the fraction of diffuse radiation caused by changes in clouds and aerosols can also interfere with our observation of the response of LUE to $R_g$ variations.

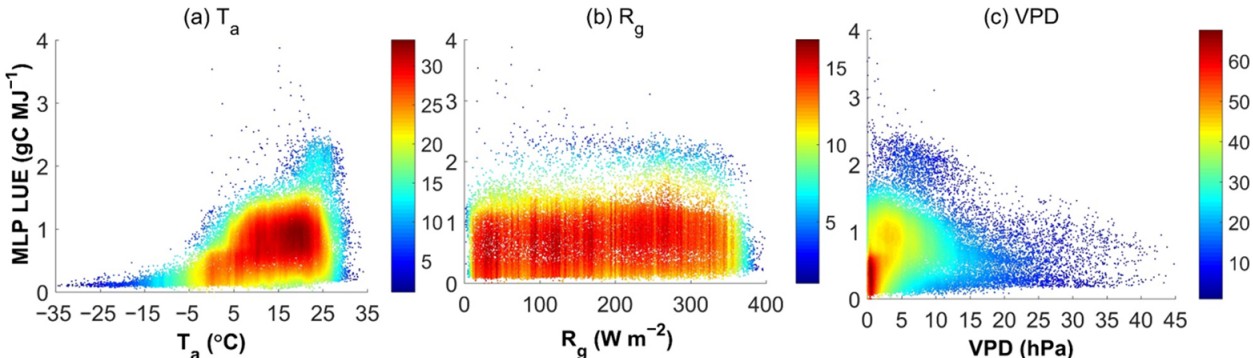

**Figure 9.** Density plot of the LUE modeled by MLP with (**a**) $T_a$, (**b**) $R_g$, and (**c**) VPD on the test dataset. The color bars represent the density of points within a circle area of radius 0.5 $^\circ$C for $T_a$, 1.0 W m$^{-2}$ for $R_g$, and 0.5 hPa for VPD.

We further investigated the responses of LUE to the three meteorological factors in different PFTs on the test dataset, as shown in Figures 10 and 11. In multiple PFTs, the LUE shows a phenomenon of increasing first and then decreasing as $T_a$ increases. For example, in DBF, EBF, ENF, GRA, MF, SAV, and SHR, the optimum temperature for vegetation photosynthesis is shown; that is, when $T_a$ reaches this value, the LUE will reach a peak, and then the LUE gradually decreases as $T_a$ increases. The global terrestrial average value of ecosystem-scale optimum temperature for photosynthesis is estimated to be $23 \pm 6\ ^\circ$C, with large spatial heterogeneity [57]. Additionally, the ecosystem-scale optimum temperature for photosynthesis varies across vegetation types [58]. In our study, for example, in ENF, when $T_a$ reached about 17 $^\circ$C, LUE reached a peak value (Figure 10(a5)), while in SAV, LUE reached a peak at 24 $^\circ$C (Figure 11(a3)). As for the response of the LUE to $R_g$ variations, in multiple PFTs such as DBF, EBF, ENF, and MF, the LUE showed a decreasing trend with an increase in $R_g$, which is caused by a light saturation effect. In most PFTs, the LUE tended to decrease as VPD increased.

In addition, it was evident that the LUE of C4 crops is higher than that of C3 crops (Figure 10). This conclusion is supported by previous studies [48,59], which found that under the same climate conditions, C4 crops had greater photosynthetic capacity than C3 crops.

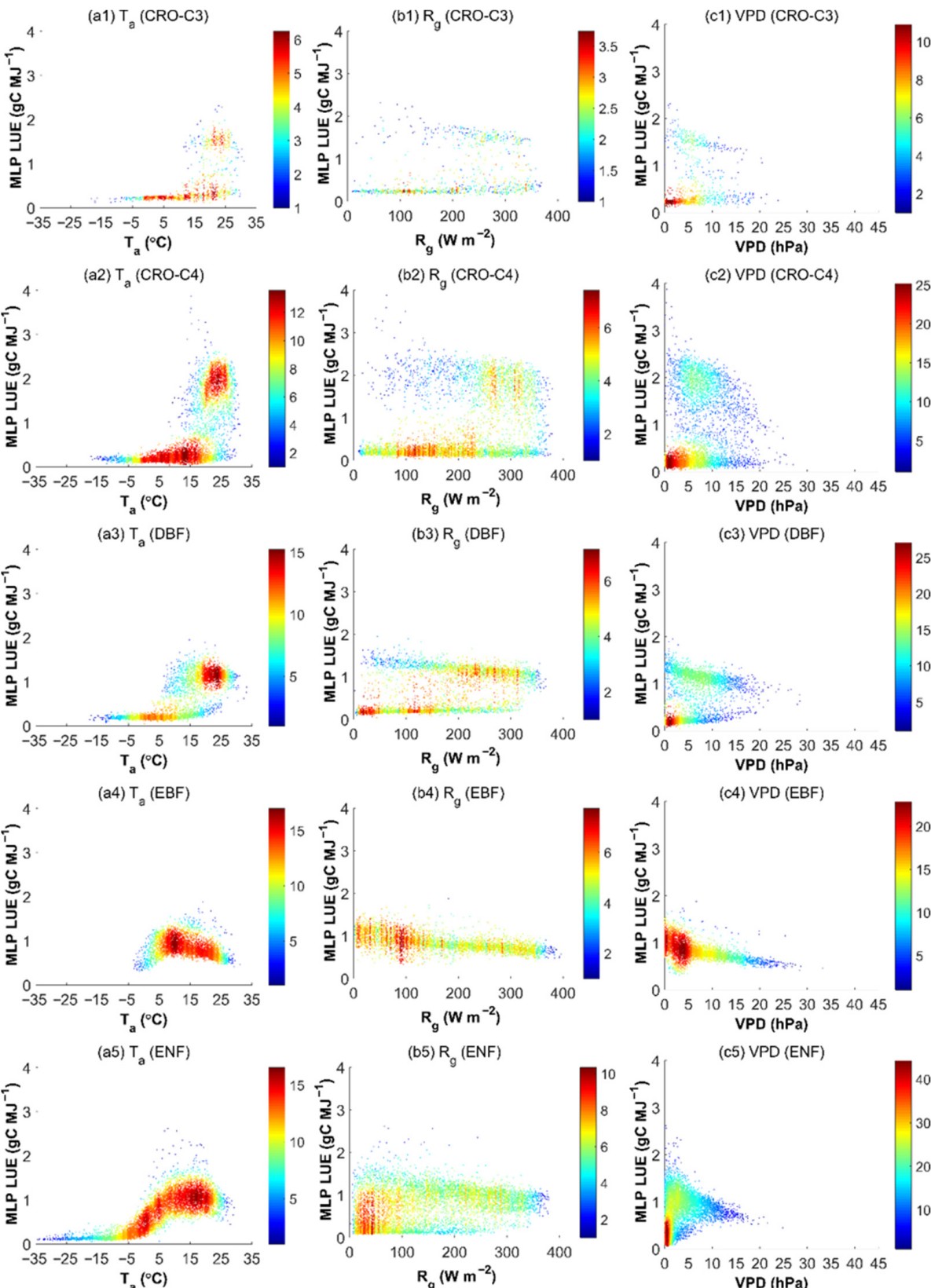

**Figure 10.** Density plot of the LUE modeled by MLP with $T_a$, $R_g$, and VPD in different PFTs on the test dataset. CRO-C3: C3 crops; CRO-C4: C4 crops; DBF: deciduous broadleaf forests; EBF: evergreen broadleaf forests; ENF: evergreen needleleaf forests. The color bars represent the density of points within a circle area of radius 0.5 °C for $T_a$, 1.0 W m$^{-2}$ for $R_g$, and 0.5 hPa for VPD.

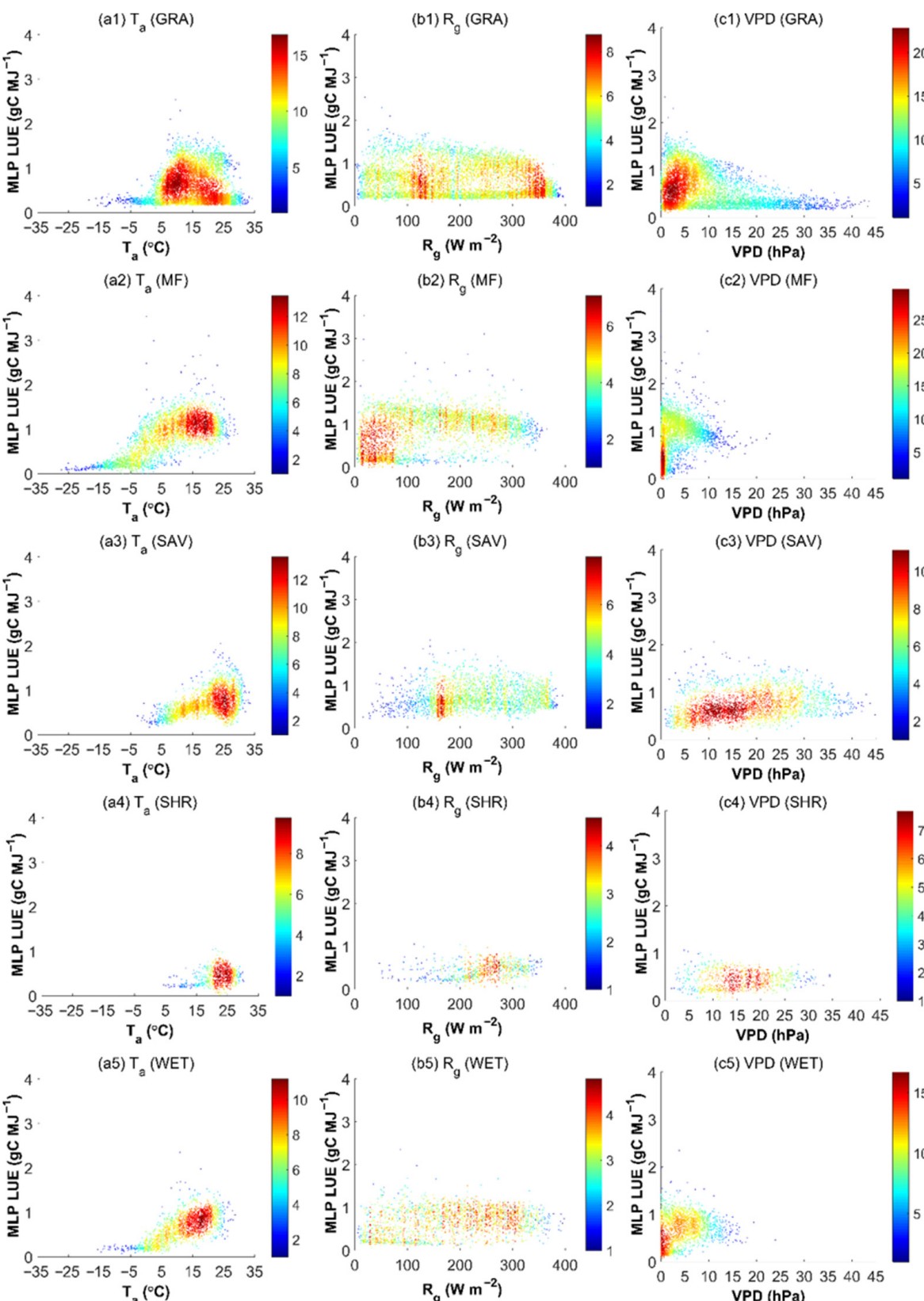

**Figure 11.** Density plot of the LUE modeled by MLP with $T_a$, $R_g$, and VPD in different PFTs on the test dataset. GRA: grasslands; MF: mixed forests; SAV: savannahs; SHR: shrubs; WET: wetlands. The color bars represent the density of points within a circle area of radius 0.5 °C for $T_a$, 1.0 W m$^{-2}$ for $R_g$, and 0.5 hPa for VPD.

## 5. Conclusions

LUE models were commonly applied to estimate the ecosystem GPP due to their concise model structures. However, quantifying $LUE_{max}$ and representing the responses of photosynthesis to environmental factors still exhibit large uncertainties, which lead to substantial errors in GPP simulations. In this study, we developed a hybrid GPP model based on machine learning and LUE models, called the hybrid light use efficiency model (H-LUE). This hybrid model was built by targeting LUE with the MLP algorithm and then, estimating GPP within the LUE model framework using the ML-based LUE and other required inputs (i.e., PAR and FPAR). We compared the performance of the H-LUE model against two widely used LUE models (VPM and VPRM) using observed daily GPP from 180 flux sites that cover nine different PFTs. The main conclusions are as follows:

1. The evaluations of the three models (H-LUE, VPM, and VPRM) when estimating the LUE and GPP indicate better performance of the H-LUE model in comparison to VPM and VPRM, which emphasizes that combining the LUE model with machine learning can lead to improved performance in comparison to conventional LUE models.

2. The H-LUE model had reasonable and significantly better performance under extremely wet, dry, high-temperature, and low-temperature conditions compared to VPM and VPRM, indicating the notable advantages of the H-LUE model for global applications. Additionally, the H-LUE model can reasonably represent the responses of photosynthesis to meteorological factors.

3. VPM and VPRM overestimate the LUE when it is small and underestimate the LUE when it is large, probably because the conventional LUE models do not account for the change in photosynthetic capacity and run with a constant $LUE_{max}$ value at all times. The H-LUE model simulates LUE using multiple input variables without using a constant $LUE_{max}$ or only two or three environmental stresses defined by a single LUE model, effectively avoiding this issue.

Overall, the developed H-LUE model can perform notably better than conventional LUE models. The new model provides a concise and effective approach to modeling the ecosystem GPP across multiple biomes and on a global scale, and it also serves as a reference for developing more advanced hybrid GPP models in the future.

**Author Contributions:** Conceptualization, D.K. and Y.B.; methodology, D.K. and Y.B.; software, D.K. and Y.B.; validation, D.K.; formal analysis, Y.B.; investigation, D.K. and Y.B.; resources, Y.B.; data curation, D.K. and Y.B.; writing—original draft preparation, D.K.; writing—review and editing, S.Z., D.Y., H.L., J.Z., S.Y., Y.L. and Y.B.; visualization, D.K. and Y.B.; supervision, S.Z.; project administration, Y.B.; funding acquisition, S.Z., S.Y., Y.L. and J.Z. All authors have read and agreed to the published version of the manuscript.

**Funding:** This research was funded by the National Natural Science Foundation of China (Grant Nos. 42101382, 42201407, and 32001130), the "Taishan Scholar" Project of Shandong Province (Grant No. TSXZ201712), and the Natural Science Foundation of Hebei Province, China (Grant No. C2021402011).

**Data Availability Statement:** The data used in the study can be downloaded through the corresponding link provided in Section B.1.

**Acknowledgments:** The authors would like to thank the editor and all anonymous reviewers for their valuable comments and helpful suggestions.

**Conflicts of Interest:** The authors declare no conflict of interest.

## Appendix A

**Table A1.** List of abbreviations and variables.

| Abbreviations or Variables | Description |
| :---: | :---: |
| $C_a$ | Carbon dioxide mole fraction |
| EnKF | Ensemble Kalman filter |
| EVI | Enhanced vegetation index |
| FPAR | Fraction of absorbed photosynthetically active radiation |
| GPP | Gross primary productivity |
| GVMI | Global vegetation moisture index |
| H | Hidden layer |
| I | Input layer |
| LUE | Light use efficiency |
| MLP | Multi-layer perceptron |
| NDDI | Normalized difference drought index |
| NDVI | Normalized difference vegetation index |
| NDWI | Normalized difference water index |
| $NIR_V$ | Near-infrared reflectance for vegetation |
| P | Accumulated precipitation value of 8 days |
| $P_a$ | Atmospheric pressure |
| PAR | Photosynthetically active radiation |
| PFT | Plant function type |
| $R_g$ | Incoming shortwave radiation |
| $R_L$ | Incoming longwave radiation |
| $T_a$ | Air temperature |
| VPD | Vapor pressure deficit |
| WS | Wind speed |

## Appendix B

*Appendix B.1 Description of VPM Model*

In the vegetation production model (VPM), $LUE_{max}$ is impacted by temperature, land surface moisture, and leaf phenology. Here is a concise overview of the VPM model:

$$GPP = PAR \times FPAR \times LUE_{max} \times T_s \times W_{SLSWI} \times P_s \qquad (A1)$$

The current version of the VPM model assumes that FPAR is directly proportional to EVI, with a fixed coefficient value of 1.0 [6]. $T_s$, $W_{SLSWI}$, and $P_s$ are scalar values that represent the impact of temperature, water, and leaf phenology on the LUE of vegetation, respectively. At each time step, $T_s$ is calculated using the equation developed for the terrestrial ecosystem model [60] as follows:

$$T_s = \frac{(T - T_{min}) \times (T - T_{max})}{\left((T - T_{min}) \times (T - T_{max}) - (T - T_{opt})^2\right)} \qquad (A2)$$

where $T$ represents the air temperature and $T_{min}$, $T_{max}$, and $T_{opt}$ represent the minimum, maximum, and optimum air temperatures (°C) for photosynthetic activity, respectively. When the air temperature falls below $T_{min}$ or exceeds $T_{max}$, $T_s$ is assigned a value of 0. In this study, $T_{min}$, $T_{max}$ and $T_{opt}$ were set to −2.5, 44.6, and 23.4 °C, respectively, for all PFTs except C4 crops, while for C4 crops, $T_{min}$, $T_{max}$ and $T_{opt}$ were set to 7.5, 43.7, and 31.3 °C, respectively [34].

The VPM also uses the LSWI (land surface water index) [6] to account for the impacts of water stress and phenology on plant photosynthesis:

$$LSWI = \frac{\rho_{NIR} - \rho_{SWIR}}{\rho_{NIR} + \rho_{SWIR}} \qquad (A3)$$

where *NIR* denotes the 841–876 nm band and *SWIR* denotes 1628–1652 nm band. The water index was defined as follows:

$$W_{SLSWI} = \frac{1 + LSWI}{1 + LSWI_{max}} \tag{A4}$$

where $LSWI_{max}$ is the highest LSWI value during the plant growing season for each pixel, $P_s$ is included to represent the impact of leaf phenology (leaf age) on photosynthesis at the canopy level, and $P_s$ is computed as a linear function of LSWI from bud burst to full leaf expansion:

$$P_s = \frac{1 + LSWI}{2} \tag{A5}$$

Once the leaves reach full expansion, $P_s$ is assigned a value of 1.

*Appendix B.2 Description of VPRM Model*

The formulation of the VPRM is based on the VPM model developed by Xiao et al. [6], which estimates the GPP using satellite-based vegetation indices and environmental data, adding a nonlinear function to take into account the response of GPP to light. VPRM can be expressed as follows:

$$GPP = PAR \times FPAR \times \frac{1}{(1 + PAR/PAR_0)} \times LUE_{max} \times T_s \times P_s \times W_{SLSWI} \tag{A6}$$

where $PAR_0$ denotes the half-saturation value. The other input variables were computed using the same method as the VPM model.

*Appendix B.3 Description of EC-LUE Model*

Yuan et al. [3,4] developed the eddy covariance-light use efficiency (EC-LUE) model to simulate the daily vegetation GPP. The EC-LUE model can be represented as follows:

$$GPP = PAR \times FPAR \times LUE_{max} \times Min(T_s, W_{SEF}) \tag{A7}$$

$$FPAR = 1.24 \times NDVI - 0.168 \tag{A8}$$

$$W_{SEF} = \frac{LE}{R_n} \tag{A9}$$

where $LUE_{max}$ is the maximum light use efficiency without environmental stress and $T_s$ is calculated using Equation (A2). If the air temperature falls below $T_{min}$ or increases beyond $T_{max}$, $T_s$ is set to 0. In this study, $T_{min}$ and $T_{max}$ were set to 0 and 40 °C, respectively, while $T_{opt}$ was determined using nonlinear optimization to be 21 °C [3]. Min denotes the minimum values of $T_s$ and $W_{SEF}$, and this model assumed that the impacts of temperature and moisture on the LUE follow Liebig's Law (i.e., LUE is only affected by the most limiting factor at any given time). *LE* is latent heat flux, and $R_n$ is net radiation.

## Appendix C

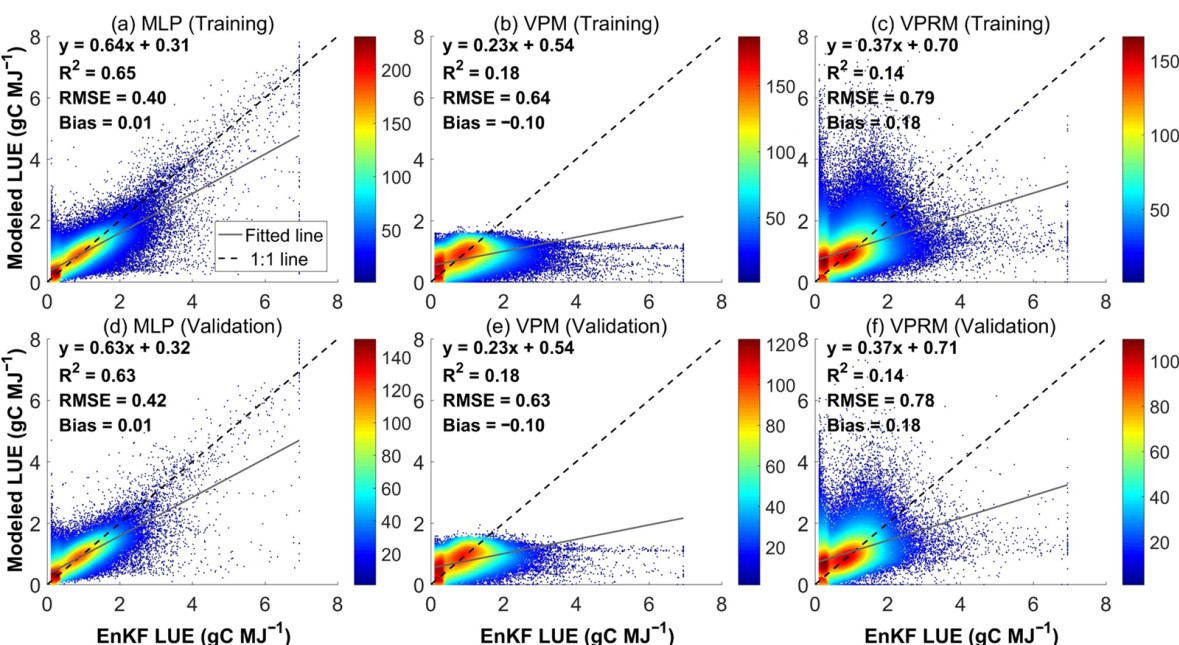

**Figure A1.** Modeled light use efficiency (LUE from MLP, VPM, and VPRM) vs. LUE obtained by EnKF for the training and validation dataset. The color bars represent the density of points within a circle area with a radius of 0.25 gC MJ$^{-1}$.

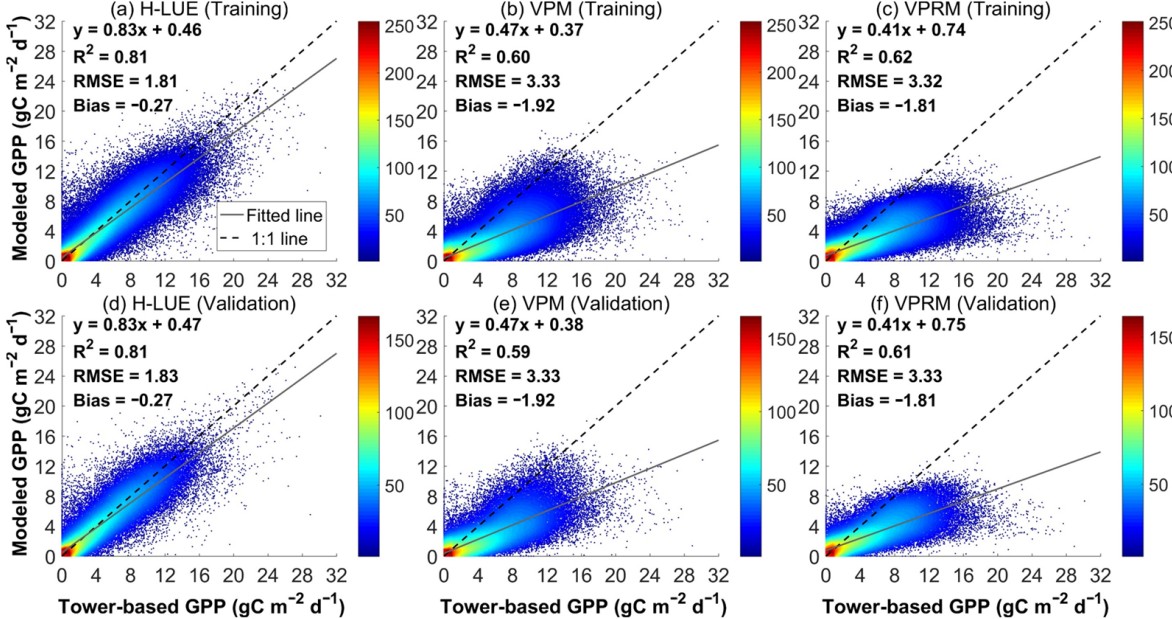

**Figure A2.** Evaluation of three models (H-LUE, VPM, and VPRM) when estimating daily gross primary productivity (GPP) on the training and validation dataset. The color bars represent the density of points within a circle area with a radius of 1.0 gC m$^{-2}$ d$^{-1}$.

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
