# Peer review of "Improving the Estimation of Gross Primary Productivity across Global Biomes by Modeling Light Use Efficiency through Machine Learning"

_remotesensing, doi:10.3390/rs15082086_

Round 1

Reviewer 1 Report

This paper developed a machine learning based-GPP model, the idea is new. However, some expressions of this paper are confusing and unclear. For example, (1) Whether the quality screening of GPP is reliable (50% maybe insufficient); (2) What are the details for data processing, for example calculating the accumulated precipitation in 8 days, and the NDVI larger than 0.2? (3) KF-based estimation of LUE should contained in the flow chart. Additionally, the author should consider whether the results of training and validation are necessary in figures 2& 3, because it only tells the readers that your whole training process ended at such a precision. (4) The author should test the performances of H-LUE in different vegetation types. (5) The discussion was insufficient, and the argument did not well support the core of the paper.

Minor comments:

L26-28: Are the R2 (0.81 – 0.86) and RMSE (1.79 – 1.83 gC m-2 d-1) according to test, training and validation? If yes, is this reasonable?

L61: “……stresses on the light use efficiency”, maximum light use efficiency?

L297-298: Whether the sequence of High-P and low-P is appropriate?

Reviewer 2 Report

Quantification of the vegetation’s primary productivity is critical to understanding the carbon cycle of terrestrial ecosystems and supporting sustainable development in remote sensing and ecological communities. The authors developed a light use efficiency (LUE) model using multi-source input variables and machine learning. They compared the new proposed model against two LUE models (VPM and VPRM) using tower-based observations. It is helpful for us to further understand the differences between model structure, LUE, and LUEmax across biomes and geographic regions.

In general, the content of this study is certainly innovative, I think that it fits within the scope of the RS journal, however, I have some comments before its consideration for publication.

1. My first concern is that VPRM and VPM are one class of models. They shared similar model structures and you also mentioned that they used the same FPAR in Lines 328-L329. Therefore, it is suggested that the authors add another LUE model, such as CASA or EC-LUE as mentioned in Introduction.

2. My second concern is the lack of evidence for your data processing and model parameters. In general, I find the paper poorly embedded in current literature. In the Introduction, the authors should make a better effort to indicate what has been done; what was the current problems to be solved in this study, and the innovation of this study. It would be helpful to illustrate the mechanism of the difference between your model and other models through data analysis and further model comparisons. This can enhance the innovation of this study.

3. My third concern is that the Discussion lacks an in-depth explanation of the innovation of your research and a comparison with the existing research results. This part needs to be further revised. Moreover, the innovation of this study mainly lies in the use of LUEmax, which is variable with seasonality, and the author needs a more in-depth analysis of this in Results and Discussion in which the current version is not sufficient.

4. The authors have given too little information about the GPP model they used. For example, how did you calculate the actual environmental stress factors (LUE) parameter, and did you keep the same input variables and model parameters as the model to be compared (VPM, VPRM)? The author must introduce these in detail.

5. English writing and clarity of writing need to improve substantially. I listed just a few edits, but major editing is needed.

Minor comments:

Abstract

L40-42: Too long. It can be split into two sentences.

Introduction

L41-42: I’m not sure how this statement shows that accurate GPP simulation can improve the soil carbon cycle.

L52-56: Here the authors need to clarify these three concepts: potential LUE, LUEmax, and actual LUE.

L67: A few pieces of literature should be added here.

L102-106: This paragraph appears unexpectedly, and the authors do not address the progress and shortcomings of the LUE model in extreme environments and the author’s comments on it.

Materials and Methods

L121: frequency->number?

L131-139: These are a little verbose and unclear.

L144: NIRV->NIRv based on Fig. 1.

L152: Remote sensing data requires additional preprocessing, including quality control such as cloud/shadows/snow removal, gap-filling, data smoothing, etc., which should be explained in more detail.

L155: unavailability.

L113: pixels-signials?

L160-164: I know the detailed methods will be explained later, but they do not seem to be well understood.

L168-187: References should be added

L188-191: The selection of parameters is too arbitrary, and the basis for parameter setting should be increased.

L195-L196: Unclear, rephrased it.

L198: What do you mean by saying dummy variables?

L218: Rg should be given full name when first appeared. The threshold of 0.5 need a basis

L221: Figure 1 is too simple to see the whole framework of the research and your innovation.

L234-L236: The original studies developed these two models (VPM, and VPRM) need to be referenced.

L243-244: P?

L246: unit.

L241-246: This section 2.2.6 is suggested to give more explanation to increase the evidence, such as an explanatory figure. Because this part is an important result.

Results

This section is not complete enough.

Figure 4: Insitu GPP observations could be added to validate these three models for modeling GPP in four extreme environmental conditions.

L247: I did not see the sensitivity analysis of the parameters, or the assessment of the impact of different parameter selections on the results, especially for the two models Ensemble Kalman Filter and Multi-layer Perceptron.

Discussion

L318-324: More mechanistic explanations for the poor performance of these two models (VPM and VPRM) are warranted. As I know, the VPM model did not fix environmental stress factors.

L337: Why these three datasets (test, training, and validation dataset) are used in Section 4.1 but only test data from these three datasets were used in Section 4.2 should be clarified.

L323, 354: The multiple input variables need to be clarified.

Conclusions

Line 359: It should be that a variable LUEmax based on machine learning can improve model performance.

Round 2

Reviewer 1 Report

The revised has been much improved. But therein lies a small problem: why you accumulated precipitation value of 8 days, rather than 5 days or 10 days?

Reviewer 2 Report

  • All concerns have been addressed.

Author Response

This paper has undergone English language editing by MDPI.

Thank you very much for your helpful comments.